# The Association between Esports Participation, Health and Physical Activity Behaviour

**DOI:** 10.3390/ijerph17197329

**Published:** 2020-10-08

**Authors:** Michael G Trotter, Tristan J. Coulter, Paul A Davis, Dylan R Poulus, Remco Polman

**Affiliations:** 1Faculty of Health, School of Exercise and Nutrition Sciences, Health Faculty, Queensland University of Technology, Brisbane QLD 4059, Australia; tristan.coulter@qut.edu.au (T.J.C.); d2.poulus@qut.edu.au (D.R.P.); remco.polman@qut.edu.au (R.P.); 2Department of Psychology, Umeå University, 901 87 Umeå, Sweden; paul.davis@umu.se

**Keywords:** BMI, exercise, video gaming, alcohol use, smoking behaviour

## Abstract

We investigated the association between obesity, self-reported physical activity, cigarette smoking, alcohol consumption, and perceived health in esports players, and the influence of player in-game rank. Data was collected with an online survey with an international participant sample of esports players representing five esports and all skill levels (*n* = 1772). Esports players were more likely to be categorized as normal weight, or obesity class 2 and 3 and as non-smokers (92%) and non-drinkers (65.1%) compared to international reference data. Esports players met international physical activity guidelines less than global general population. Esports players ranked in the top 10% were more physically active compared to the remaining esports players. As esports player in-game rank increased, so did the amount of time spent playing esports. Although esports players appear generally healthy, a small group was significantly obese and most esports players did not meet physical activity guidelines, indicating potential future health risks.

## 1. Introduction

Esports has been defined as “an organized and competitive approach to playing computer games” [1]. The scale of esports competitions can range from small competitions organized between friends to international events with millions of dollars in prize money at stake [2]. In 2020, the esports industry has reached a value of more than $1 billion USD and been projected to grow at a rate of over 15.7% per year [3]. All esports are video games, but not all video games are esports. The defining feature of esports is the competitive nature of the video game. Esports are video games specifically designed with competition in mind. Video games, on the other hand, are a leisure activity designed to entertain [4]. Esports competitions at the highest level often have live events, cash prizes, and online/in-person viewership. Although extensive research has been conducted into video gaming in relation to health and physical activity (PA) behaviour, to date, limited studies have examined this relationship in esports.

### 1.1. Sedentary Behaviour and Body Mass Index

An important issue for public health is whether the accelerated growth in esports is accompanied by increased time spent sitting while playing esports, as non-active video games have been associated with an increase in Body Mass Index (BMI) [4]. Increased screen time is associated with higher levels of obesity and unhealthy lifestyle behaviours in adults [4,5,6]. Despite increased screen time being linked to increased rates of obesity, the link between video gaming and obesity has not been strongly established [4]. Previous research on the health and weight status of esports players has indicated the need for health-promotion strategies for esports players, due to the relationship between sedentary lifestyles and poor health of esports players [7].

Empirical evidence suggests that excessive video gaming and increased screen time are associated with higher obesity levels in adults [4,6,8], especially abdominal obesity [9]. Previous research has shown that an increased amount of time spent video gaming on the weekend was associated with increased risk of being obese [5], as well as being less likely to meet World Health Organization (WHO) PA guidelines [10]. The association between video gaming and abdominal obesity [9] is particularly worrying, since abdominal obesity is independently related to negative cardio-metabolic health consequences in adults [11,12]. It has been shown that higher levels of video game addiction were associated with increased abdominal obesity [9], due to disturbed sleeping patterns. Increased screen time in adults has also been associated with higher BMI, a finding shown to be independent of PA behaviour [13]. Overall, there appears to be a positive correlation between sedentary lifestyles and obesity [13,14,15]. Limited research has investigated the relationship between esports participation and BMI. Only one study [7] has directly examined esports BMI status, and found that the average esports players BMI in Germany was classed as normal weight. Further research is required to understand what BMI esports players have globally and how esports players compare to population reference data.

### 1.2. Physical Activity Behaviour

Anecdotal evidence suggests that esports players are sedentary for 4.2 h each day while training [16]. The amount of time esports players spend sitting has potential negative consequences, including a higher risk of injuries and chronic diseases, such as upper extremity dysfunction, metabolic dysregulation, circadian rhythm problems, and neck and back problems [17]. To date, no empirical evidence exists about whether esports players are more or less physically active than general populations and if this relationship is associated with other negative health behaviours and obesity levels.

Previous research has shown that esports athletes’ PA levels exceed the World Health Organization’s PA guidelines [10,16,18]. It has been reported that as part of their training regime, elite esports athletes spend 1.08 h per/day engaging in physical exercise [16]. Anecdotal evidence also suggests that esports athletes include physical training as a strategy to enhance gameplay and manage stress [19]. However, only 6–9% of esports players report engaging in exercise for the perceived performance benefits, whereas 32–47% engage in PA primarily for general health benefits [16,18]. Despite both individual and team esports athletes reporting PA as a component of their overall training, there is currently no research indicating what effect this activity has on players’ overall performance, or their general health, including obesity levels.

### 1.3. General Health

The association between general health and video games has received extensive attention in the academic literature [20,21,22], whereas the association between esports participation and general health has not [7]. Often in video gaming studies, general health is measured through a combination of physical, mental, and/or social health factors [21,23]. Findings indicate that negative health outcomes (e.g., mental health, general health SF-1 and PA) are associated with problematic use of video games (e.g., gaming addiction—salience, tolerance, mood modification, withdrawal, relapse, conflict and problems) [24]. An increase in video game playing time has been associated with poor general health [25]. However, to date, limited research has been conducted examining the effect of esports involvement on general health.

### 1.4. Health Behaviours

Lifestyle behaviours, including alcohol consumption and cigarette smoking, predict higher rates of mortality [26]. In addition, PA behaviour has been found to be ineffective in compensating for the detrimental effects of smoking [27]. Video gaming has been associated with increased substance abuse, with a linear relationship being found between average video game play time and the use of alcohol and drugs (including nicotine) in Norwegian adults [25]. Alcohol use is also a risk factor for obesity [28], whereas the relationship with smoking is more complex. For example, in a large cross-sectional study, smoking was not found to be protective against obesity in younger and heavier smokers. However, smokers were less likely to be obese than people who had never smoked [29]. Regarding alcohol consumption, moderate drinkers have been found to be more likely to be physically active [30,31]. However, no study is yet to examine alcohol and smoking frequency in esports players and their association with PA and obesity in this population. Currently, no study has compared the health behaviours of esports players with international population reference data.

### 1.5. Study Aims

This study’s first aim was to investigate the relationships between obesity (BMI), and self-reported PA levels, drinking and smoking frequency, and general health perceptions in esports players. A second aim is to investigate the cross-sectional association between BMI, physical activity, general health (SF-1), smoking and drinking frequency and player in-game rank. The final aim of this study is to compare esports players’ BMI and smoking and drinking frequency to international reference data to see how esports players compare with global populations. It is predicted that esports players will report greater BMI than general populations due to the sedentary nature of esports, and predict that esports players will drink and smoke more and report lower PA levels than the general populations. Finally, it is predicted that esports player in-game rank will inversely correlate with BMI and be positively associated with esports frequency, PA frequency, and general health (SF-1), and negatively associated with smoking and drinking behaviour.

## 2. Materials and Methods

### 2.1. Participants

In total, 2459 participants completed the study survey. All participants provided informed consent prior to starting the survey. However, only 1772 individuals (72.1%) completed sufficient and reliable information for data analysis. Less than half of participants (47.6%, *n* = 843) provided information about where they lived and their gender (male = 87.2% *n* = 742, female = 9.4% *n* = 80, other = 3.4% *n* = 29). Participants in this study lived in 65 countries (*n* = 843); the five largest groups included the United States of America (34.4%, *n* = 290), Australia (21.4%, *n* = 180), Canada (8.9%, *n* = 75), Germany (4.9%, *n* = 41), and the United Kingdom (3.1%, *n* = 26). Table 1 shows the frequencies of participants for the different countries of residence. All participants self-selected to take part in this study when they clicked the link promoted through social media. Participants at live esports events were selected by convenience sampling, where players who walked past the researchers were approached and asked to take part in the survey or handed a flyer with a link to the survey.

### 2.2. Measures

General health: The SF-1 was used to measure general health [32]. The SF-1 is the short version of the commonly used SF-36, which measures general health perception [32,33]. The SF-1 includes one item: “In general, would you say your health is?” and is scored on a 5-point Likert scale (Excellent, Very good, Good, Fair and Poor). The SF-1 is a valid and reliable measure of self-reported health which has been used by the South Australian Government [32] and the Australian Bureau of Statistics [34] and predicts health behaviours, such as PA, smoking, alcohol consumption, and weight status [35], as well as mortality and morbidity [36].

Drinking and smoking: Following previous research [37], this study investigated smoking and drinking behaviour by using a single item for smoking and drinking frequency. These items have been adapted from previous research (“Have you ever smoked a cigarette?” and “Have you ever drank alcohol” to “How many days per week do you engage in smoking? How many days per week do you engage in drinking?). Scores were adapted to represent days per week with scores ranging from 0 days to 7 days, instead of yes or no. Single question measures of days per week consuming alcohol [38] and cigarettes [39] have been previously used to measure these behaviours in adult populations.

Body Mass Index (BMI): BMI (kg/m^2^) was assessed using self-reported height and weight. In accordance with [40], BMI categories are as follows: <18.5 = underweight, 18.5–24.9 = normal weight, 25–29.9 = pre-obesity, 30.0–34.9 = obesity class 1, 35–39.9 is obesity class 2, >40 = obesity class 3 [40]. BMI has been adjusted by 5.8% to account for self-report bias in height and weight in this study, based on findings from previous research, which suggest that participants underestimate their BMI [41].

Physical activity (PA): PA was measured to identify perceived PA level and days physically active per week. Perceived PA level was measured with one item: “In general, how physically active would you say you are?”. Days physically active per week was measured with one item: “How many days per week do you participate in sport or physical activity for a total of 30 min or more per day?” This item has been validated and used for the Active Australian PA recommendations and has been shown to be a valid and reliable correlate of objectively measured PA [42,43] and Australian PA recommendations for 12–18 year olds [44]. Days physically active per week was also measured using one item [43]: “How many days per week do you participate in sport or physical activity for a total of 60 min or more per day?”. This study reduced the total minutes per day to 30 min to determine how many esports players were meeting the minimum levels of PA set by the WHO PA guidelines [10]. Participants who reported exercising a minimum of 30 min a day, at least 5 days a week, were considered to meet the WHO PA guidelines [10].

Player in-game rank: Esports games categorize players into skill groups determined by an in-game algorithm, which uses in-game data including (but not exclusively) numbers of wins and losses [45]. Each game skill groups do not align with the other games skill groups; therefore, to standardize scores, players were allocated into 4 categories based on skill group: category 1 = 0–69%, category 2 = 70–79%, category 3 = 80–89%, category 4 = 90–100%. Previous research on esports performance has used this same method for categorizing esports performance [46].

Esports frequency: This study measured esports engagement in terms of number of days per week spent playing esports: “How many days per week do you play video games competitively?” This measure has been validated previously when examining active and problematic video gaming, which measured days per week to determine the characteristics of video game use [47,48]. The present study added the word ‘competitively’ at the end of the question to align it to previous definitions of esports [1].

Software: Data was collected through the survey platform Qualtrics (Qualtrics, Provo, UT, USA). Population data was provided by NCDRisC in excel (Microsoft, Redmond, WA, USA). Data analysis was conducted IBM SPSS (Statistical Package for the Social Sciences) Statistics 26 (IBM, Armonk, NY, USA).

### 2.3. Procedure

This study was approved by a University’s Research Ethics Committee. The Queensland University of Technology ethics approval number is 1800000436. Participants provided informed consent prior to participating. Participants were recruited through two methods: (i) direct contact at a major esports event in Australia, and (ii) social media advertisement. Prior approval was obtained from event organizers before obtaining data at a major esports event. The researchers approached potential participants and explained the study aims. If participants volunteered to take part in this study, they completed the survey via a link to the online portal on an iPad provided by researchers (all participants completed the survey electronically). Otherwise, participants accessed the survey through a weblink advertised through social media, which took approximately 20 min to complete.

### 2.4. Analysis Strategy

Data was screened for outliers, and unrealistic or spurious data was removed. Following this, descriptive statistics were obtained for all variables and Pearson Product Moment correlations were calculated between all study variables. To examine our first aim (to investigate the relationship between BMI, self-reported PA levels, drinking and smoking frequency and general health perceptions in esports players), this study used multivariate analysis of variance (MANOVA). BMI category was the independent variable and the dependent variables were perceived PA level, PA, smoking and drinking frequency, and esports frequency. To assess the results of the second aim of this study (to investigate the cross-sectional association between BMI, physical activity, general health (SF-1), smoking and drinking frequency and player in-game rank), MANOVA was conducted with player in-game rank as the independent factor and BMI categories, general health (SF-1), perceived PA level, PA, smoking and drinking frequency and esports frequency as the dependent variables. To determine the results of the final aim of this study (to compare esports players’ BMI and smoking and drinking frequency to international reference data to see how esports players compare with global populations), a chi-square analysis was used to compare the population reference data for global BMI category percentages against the sample BMI category percentages. Observed and expected participant numbers, based on the NCDRisC data [49], are reported along with percentages of each BMI category. In the instance of a significant main effect in the MANOVA, univariate analysis of variance (ANOVA) was conducted. Significant ANOVA effects were followed up with Sidak post-hoc comparisons. Effect sizes for ANOVAs are reported as partial eta squared (η_p_^2^), with a small effect at 0.01–0.059, a medium effect at 0.06–0.139, and a large effect at >0.14. Effect sizes for correlations are reported as Pearson’s *r*, with a small effect at 0.10–0.29, a medium effect at 0.30–0.49, and a large effect at >0.50 [50].

## 3. Results

We first examined the normality and homoscedasticity of our dataset. Although the Kolmogorov–Smirnov test showed deviation from normality for the variables of this study (*p* < 0.05), it is well established that in large sample sizes (>40) the violation of this assumption has little influence and parametric statistics can be performed [51]. There was no violation with regard to homoscedasity.

### 3.1. Frequencies, Means, and Standard Deviations

Frequencies for the health behaviour variables are illustrated in Table 2 and Table 3. Table 4 provides an overview of the Pearson Product Moment correlations between study variables. As predicted, player in-game rank was moderately and positively associated with esports frequency, and weakly associated with perceived PA level and inversely correlated with BMI (i.e., better players had a lower BMI). In addition, esports frequency was associated with higher levels of perceived PA level, and general health (SF-1). Drinking frequency was associated with higher smoking frequency and higher perceptions of PA level. General health (SF-1) was moderately and positively associated with perceived PA level. Inversely, general health (SF-1) was negatively associated with BMI and with smoking frequency. Finally, both perceived PA level and PA were inversely associated with actual BMI.

### 3.2. BMI and Population Reference

The chi-square goodness-of-fit test indicated that there were significant differences in the proportion of participants in each BMI category compared to global population data (ꭓ^2^(5, *n* = 831) = 113.86, *p* < 0.001). There were also significant differences when comparing the sample of esports players from their respective countries. For instance, players from the United States differed to the general USA population (ꭓ^2^(5, *n* = 285) = 76.41, *p* < 0.001) and Australian esports players differed from the Australian general population (ꭓ^2^(5, *n* = 176) = 45.49, *p* < 0.001). No other county had a large enough sample to have a player in each BMI category and therefore was not analysed. Results for the chi-square analysis are shown in Table 5.

### 3.3. PA, Smoking, and Drinking Frequency

In terms of days physically active per week, 19.7% of esports players met the WHO guidelines by participating in PA for a minimum of 30 min at least 5 days per week. For smoking frequency, 3.7% (*n* = 66) of the esports players smoked daily, 8% (*n* = 141) at least once a week and 92% (*n* = 1622) reported to not smoke at all. Esports players who smoked cigarettes, smoked on an average of 4.56 days per week. 0.5% (*n* = 9) Of esports players reported drinking daily, whereas 34.9% (*n* = 616) reported drinking alcohol at least once a week. Overall, 65.1% (*n* = 1150) of esports players reported not to drink at all. Esports players who drank alcohol averaged drinking on 1.80 days per week.

### 3.4. BMI and Health Behaviours

The MANOVA for BMI category was significant (Wilk’s lambda = 0.82, *p* < 0.001; ηp^2^ = 0.04). Follow-up ANOVA showed significant differences for perceived PA level (F(5,826) = 8.06; *p* < 0.001; ηp^2^ = 0.05), days physically active per week (F(5,826) = 3.00; *p* = 0.01; ηp^2^ = 0.02), perceived general health (F(5,678) = 24.32; *p* < 0.001; ηp^2^ = 0.15), drinking frequency (F(5,678) = 3.16; *p* = 0.01; ηp^2^ = 0.023), and player in-game rank (F(5,678) = 2.34; *p* = 0.04; ηp^2^ = 0.02), but not esports playing frequency (F(5, 678) = 0.34; *p* = 0.89; ηp^2^ = 0.003) and smoking frequency (F(5,678) = 1.24; *p* = 0.29; ηp^2^ = 0.01).

Post-hoc comparisons for perceived PA level found that esports players in obesity class 1, 2, and 3 scored significantly lower on activity levels than those in the normal and pre-obese range, but not significantly more than those in the underweight category (all *p* < 0.05). However, no significant differences were found between groups for days physically active per week. Post-hoc comparisons for perceived general health showed that esports players in obesity class 1, 2, and 3 had significantly lower general health (SF-1) scores than players classified as underweight, normal, or pre-obese (all *p* < 0.05). For drinking frequency, esports players classified as underweight drank alcohol significantly less days than players classified as normal, pre-obese, and obesity class 2 (all *p* < 0.05). Post-hoc comparisons for player in-game rank did not show any significant differences.

### 3.5. Player in-Game Rank and Health Behaviours

MANOVA exploring the role of player in-game rank on perceived activity level, days physically active per week, perceived general health, drinking frequency, smoking frequency, and esports playing frequency was significant (Wilk’s lambda = 0.89, *p* < 0.001; ηp^2^ = 0.04). Follow-up ANOVAs showed significant differences for esports playing frequency (F(3,1378) = 53.84; *p* < 0.001; ηp^2^ = 0.11) and days physically active per week (F(3,1376) = 4.71; *p* = 0.01; ηp^2^ = 0.01). However, the ANOVAs for perceived PA level (F(3,1383) = 1.65; *p* = 0.17; ηp^2^ = 0.004), perceived general health (F(3,1383) = 0.98; *p* = 0.40; ηp^2^ = 0.002), drinking frequency (F(3,1375) = 0.98; *p* = 0.40; ηp^2^ = 0.002), and smoking frequency (F(3,1372) = 0.22; *p* = 0.88; ηp^2^ = >0.001) were not statistically significant.

Post-hoc comparisons for playing frequency showed that all groups were different from each other (all *p* < 0.05). The participants ranked in the 90–100% group played the most days per week (3.98 days), followed by the 80–89% group (3.21 days), followed by the 70–79% group (2.63 days), and finally the 0–69% group played the least (2.05 days). For days physically active per week, post-hoc comparisons only showed differences between the 90–100% and the 0–69% groups (*p* < 0.05). The 90–100% group engaged in PA on average 2.79 days, whereas the 0–69% reported an average of 2.24 days.

## 4. Discussion

This study compared the BMI status, PA behaviour, smoking and drinking frequency, perceived general health, and practicing behaviour of esports players of different ability, including the associations between these factors. Results indicated that esports players’ BMI differed significantly from global BMI population reference data [49], with esports players more likely to have their BMI categorized as normal weight, pre-obese and obese class 2 and 3. In addition, a healthier BMI was associated with higher levels of perceived PA level and higher levels of perceived general health. The percentage of players ranked in the top 10% reported to be significantly more physically active than the remaining 90% of players. As esports player in-game rank increased, so did the number of days spent playing esports. However, this relationship was inversely correlated with actual BMI. Esports player in-game rank was not associated with drinking or smoking frequency or general health (SF-1).

### 4.1. BMI Esports Players in Comparison to International Population Data

This is the first study to examine the BMI status of a global sample of esports players. A previous study [7] has examined the demographics of esports players including BMI in Germany. However, previous research did not report the difference between esports players and general populations. Research [7] has shown that esports players mean BMI ranged between 23.1 and 26 kg/m^2^. This would indicate that esports players generally are a healthy weight or marginally pre-obese. In video gamers, a significant inverse relationship between video gamers BMI and the amount of time spent playing video games has been found [4]. However, only 1% of the participants’ BMI was explained by time spent gaming. Findings in the present study showed that the distribution of esports players in BMI categories were significantly different from global population reference data [49]. In this study, the distributions of the two largest subpopulations of esports players—the United States of America (34.4%) and Australia (21.4%)—also differed significantly from the respective American and Australian general populations. Globally, esports players were more likely to be categorized in normal weight and obesity classes 2 and 3, compared to international reference data, and less likely to be classed in underweight, pre-obese and obesity class 1 categories. When examining the USA and Australian esports players’ BMI, they were more likely to be of normal weight compared to their respective populations (18.07% more for USA and 21.31% for Australian sample). In addition, the USA and Australian esports players were less likely to be categorized in obesity classes 1 and 2, whereas the USA esports players were more likely to be categorized in class 3 (1.44%). These results show that globally esports players are classed as healthy weighted (9.93% more) than the global population reference data. Esports players are however, also 4.03% more likely to be morbidly obese compared to global population reference data.

There has been very limited research on esports players’ BMI. Only one study has indicated that the average player’s BMI can be classed as normal [7]. In contrast, studies examining the relationship between video game use and BMI status has been equivocal. For example, the BMI of video game players has been found to be higher compared to non-gamers [52]. However, there are claims that video gaming and other sources of media have a lessor impact on BMI, compared to socio-economic status and socio-demographic factors [53]. Other factors have also affected the association between BMI and video game play, such as length of game play in one sitting, years of video game playing, and online vs. offline games [6]. Overall, esports might be less detrimental to the BMI of the participants compared to computer gaming or other sedentary activities. Present study findings support previous reported results, where esports players have been categorized in the normal range for BMI. However, the current study also reports that the global sample of players shows a larger number of players being considered severely and morbidly obese (i.e., categorized into obesity classes 2 and 3). Findings in the present study suggest that strategies should be developed to support esports players who are classes at the higher end of BMI categories.

### 4.2. BMI and Health Behaviours

The correlational analysis showed that higher levels of actual BMI were lower and inversely associated with perceived PA level, days physically active per week, and perceived general health. In addition, there was a significant small positive correlation between smoking and drinking frequency. The esports players reported that 3.7% smoked daily and 8% at least once a week. This finding indicates that player smoking frequency appears to be lower compared to global data (average 18.7% daily smoking), although variations occur across countries [54]. For example, in Australia, 13.8% of people are daily smokers, with an additional 1.4% reporting to be smokers but not on a daily basis [55]. Similarly, in the United States, 15.6% of men and 12% of women smoke cigarettes daily or on some days [56].

In addition to having a relatively low smoking frequency (8%, *n* = 141), the majority of esports players reported not drinking alcohol (65.1%, *n* = 1150). Global data suggests that 58.2% of adults abstain from drinking alcohol [57]. Esports players are 7.8% more likely to abstain from drinking alcohol than global reference data [57]. Of those players that do drink, only 0.5% reported drinking daily, whereas 34.9% reported drinking at least once per week.

The WHO [10] has set guidelines for the amount of time should be spent being physically active weekly, recommending a minimum of 150 min of PA per week. This study shows that, as a group, 80.3% esports players are not meeting the WHO PA guidelines [10] at the same rates as others in the general population. Participants from the USA and Australia made up the largest proportion of the sample, in this study (55.8%). Approximately 78% (*n* = 225) of USA and 74% (*n* = 128) of Australian esports players did not meet PA guidelines, whereas 40% and 30.4% of the general population from these respective countries did. As esports is sedentary in nature, the overall lack of PA being undertaken by esports players could leave large numbers of players at future risk of serious health problems, such as back, neck and wrist injuries [17], as well as chronic diseases associated with sedentary behaviour [58].

As found previously, perceptions of general health had small correlations with perceived PA level, days physically active per week, and smoking frequency, but not alcohol consumption [35]. The complex association between perceptions of general health and alcohol consumption, and the notion that 65% of the sample did not consume alcohol, might be responsible for the lack of correlation between these variables.

### 4.3. Player in-Game Rank and Time Spent Playing Video Games

The a priori assumption that higher-ranked esports players would spend an increased time playing esports was supported. Superior player in-game rank was associated with more time spent playing esports. Previous research has suggested that practice time is an important factor when predicting performance outcomes [59,60].

### 4.4. Player in-Game Rank and Health Behaviours

Partly supporting our hypothesis, results showed that players in the top 10% of player in-game ranks participated in PA more days per week compared to those in the bottom 90% of esports players. Previous research investigating PA and esports players found that the motivation to participate in PA is most frequently for general health benefits, with a smaller percentage of players being physically active with performance enhancement in mind [16,17,18]. Although the present study is cross-sectional in nature, it appears that those players with a higher in-game rank are more likely to be active on more days. There is likely to be a reciprocal relationship between these two variables. On the one hand, higher levels of PA have been associated with increased cognitive functioning in areas relevant for esports performance, including spatial cognition and memory [61,62]. On the other hand, higher level esports players might have more opportunities to be physical active. In particular, at the highest-level, teams engage in organised physical training to enhance performance [19].

It was predicted that higher ranked players would engage in fewer negative health behaviours, such as smoking and drinking alcohol, and have greater reported general health compared with lower ranked players. Results indicated that no significant association between drinking frequency and smoking frequency was identified with esports player in-game rank. One potential explanation for this finding might be the low levels of drinking frequency (46.8% non-drinkers) and smoking frequency (92% non-smokers) in this study’s sample. In a recent study of high-level esports players [46], it was found that substance use was not a popular coping strategy used to deal with stress associated with competing in esports. No other studies to date have evaluated the use of alcohol and cigarette use by esports players. Future research on the use of alcohol and cigarettes would help fill a gap in esports health literature.

Finally, no significant relationship was found between general health (SF-1) and player in-game rank. This finding is in line with no reported differences in drinking frequency, smoking frequency, and perceived PA level across any of the esports ranking groups.

### 4.5. Study Limitations and Future Research

A limitation of the present study was that all the variables relied upon self-report. It is, for example, well known that individuals under report their weight, and over report their height and activity level [63]. In addition, to promote participant responses, this study predominantly used single-item measures. Although these items have been shown to have good reliability and validity in previous studies, future studies could explore aspects of health behaviours or perceptions of health (e.g., physical vs. mental health) in more detail. In addition, more objective data collection methods could be used to explore PA behaviour and actual time spend playing esports. This could include accelerometers and monitoring of time spend playing esports. Additionally, it would be particularly interesting to examine more closely the association between PA behaviour and esports performance [64]. It is still unknown what the optimum exercise dose and mode (e.g., aerobic vs. resistance) would be to achieve higher levels of esports performance, over time. Although a global sample of players was sourced, the survey was distributed only in English, this may have impacted either the understanding of participants from other countries of the questions being asked, or the likelihood of participants self-selecting to participate in this study due to a language barrier. Lastly, participants self-selected to take part in this study, which may have influenced the results of this study.

## 5. Conclusions

This study investigated (i) the association between obesity, PA, cigarette smoking, drinking frequency, and general health (SF-1) in esports players, and (ii) the role that online player in-game rank plays as a mediating factor for BMI, PA, perceived general health, smoking and drinking frequency in esports players. It was found that more esports players were classed as normal weight compared to global population data, although the number of esports players in obesity classes 2 and 3 was higher compared to general populations. Additionally, the reported rate of smoking and drinking frequency was relatively low, with 65.1% being non-drinkers and 92% being non-smokers. This study partly supported the prediction that higher-ranked esports players were more physically active, with the top 10% of players significantly more physically active than the bottom 90% of players. Neither smoking nor drinking frequency was associated with player in-game rank. Esports players would benefit from increased PA behaviour, as this might positively decrease obesity levels, as well as enhance esports performance.

## Figures and Tables

**Table 1 ijerph-17-07329-t001:** Frequency table of player country of residence.

	Countries with 1 Participant (*N*)	Countries with between 2 and 10 Participants (*N*)	Countries with between 11 and 25 Participants (*N*)	Countries with more than 25 Participants (*N*)
Country name	American Samoa (1), Aruba (1), Bahrain (1), Bangladesh (1), Bosnia and Herzegovina (1), British Virgin Islands (1), Bulgaria (1), China (1), Greece (1), Guatemala (1), Hong Kong (1), Iceland (1), Korea (1), Kosovo (1), Latvia (1), Luxembourg (1), Malaysia (1), Norway (1), Puerto Rico (1), Singapore (1), Slovenia (1), Taiwan (1), Ukraine (1), Viet Nam (1)	Argentina (7), Austria (4), Belgium (4), Chile (4), Colombia (2), Croatia (2), Czech Republic (4), Denmark (8), Estonia (2), Finland (6), Hungary (5), India (3), Indonesia (3), Ireland (8), Israel (2), Italy (5), Japan (2), Lithuania (2), Mexico (8), New Zealand (3), Peru (3), Philippines (6), Portugal (3), Romania (4), Russia (4), Serbia (4), South Africa (4), Spain (4), Switzerland 4), Uruguay (2)	Brazil (12), France (17), Netherlands (15), Poland (15), Saudi Arabia (17), Sweden (17), United Kingdom (26)	Australia (180), Canada (75), Germany (41), United States of America (290)
Number of Countries	24	30	7	4

**Table 2 ijerph-17-07329-t002:** Mean and standard deviation for all factors by player in-game rank category.

Variable		Esports Frequency	Drinking Frequency	Smoking Frequency	Perceived PA Level	General Health (SF-1)	PA (Days Physically Active *p*/w)	BMI
0–69%	M	2.36	0.68	0.38	2.45	3.08	2.30	27.00
	SD	2.06	1.19	1.46	0.856	0.949	1.98	8.11
	*N*	360	360	360	360	360	360	360
70–79%	M	2.75	0.69	0.38	2.51	3.14	2.72	26.21
	SD	2.05	1.20	1.41	0.817	0.890	2.25	6.85
	*N*	102	102	102	102	102	102	102
80–89%	M	3.62	0.51	0.38	3.12	3.12	2.47	25.11
	SD	2.08	0.90	1.44	0.914	0.971	2.23	7.00
	*N*	97	97	97	97	97	97	97
90–100%	M	4.06	0.59	0.37	2.55	3.09	2.81	25.26
	SD	2.16	1.05	1.51	0.923	0.980	2.09	6.74
	*N*	124	124	124	124	124	124	124

*p*/w = per week; PA = physical activity.

**Table 3 ijerph-17-07329-t003:** Drinking and smoking response frequency.

Days per Week	Drinking Frequency	Smoking Frequency	PA
0	1150	1622	440
1	372	35	300
2	126	13	278
3	53	9	263
4	26	5	137
5	22	5	191
6	8	8	66
7	9	66	92

**Table 4 ijerph-17-07329-t004:** Correlations between all factors.

Variable		Esports Frequency	Drinking Frequency	Smoking Frequency	Perceived PA Level	General Health (SF-1)	PA (Days Physically Active *p*/w)	BMI
Player in-game rank	r	0.32 ***	−0.04	−0.001	0.06 *	0.04	0.082 **	−0.11 **
	*N*	1379	1376	1384	1384	1384	1377	690
Esports frequency	r		0.02	0.01	0.10 ***	0.11 ***	0.13 **	−0.01
	*N*		1762	1759	1772	1772	1765	826
Drinking frequency	r			0.27 ***	0.09 ***	0.04	0.04	0.01
	*N*			1762	1766	1766	1761	824
Smoking frequency	r				0.02	−0.07 **	0.03	0.06
	*N*				1763	1763	1758	831
Perceived PA Level	r					0.47 ***	0.59 ***	−0.08 ***
	*N*					1781	1767	831
General health	r						0.27 ***	−0.29 ***
	*N*						1767	831
PA	r							−0.08 *
	*N*							826

Pearson Product Moment correlations between study variables (* *p* < 0.05; ** *p* < 0.01; *** *p* < 0.001). *p*/w = per week; PA = physical activity.

**Table 5 ijerph-17-07329-t005:** Results of the chi-square test of independence between observed BMI of esports sample and population data obtained from the NCDRisC [49] global population reference data.

Country	Chi-Square Test	Observed *N*	Expected *N*	Residual	Observed Sample %	NCDRisC Population %	Percentage Difference
Global	Underweight	60	97.7	−37.7	7.22	11.76	−4.54
Global	Normal weight	376	293.5	82.5	45.25	35.32	9.93
Global	Pre-obese	201	259.2	−58.2	24.19	31.19	−7.00
Global	Obese class 1	95	119.9	−24.9	11.43	14.43	−3.00
Global	Obese class 2	46	41.1	4.9	5.54	4.95	0.59
Global	Obese class 3	53	19.5	33.5	6.38	2.35	4.03
Global	Total	831					
Australia	Underweight	9	6.2	2.8	5.11	3.52	1.59
Australia	Normal weight	86	48.5	37.5	48.86	27.56	21.31
Australia	Pre-obese	43	67.8	−24.8	24.43	38.52	−14.09
Australia	Obese class 1	20	34.3	−14.3	11.36	19.49	−8.13
Australia	Obese class 2	12	13.2	−1.2	6.82	7.50	−0.68
Australia	Obese class 3	6	6	0	3.41	3.41	0.00
Australia	Total	176					
USA	Underweight	21	9.4	11.6	7.37	3.30	4.07
USA	Normal weight	124	72.5	51.5	43.51	25.44	18.07
USA	Pre-obese	62	98.8	−36.8	21.75	34.67	−12.91
USA	Obese class 1	35	57	−22	12.28	20.00	−7.72
USA	Obese class 2	19	27.3	−8.3	6.67	9.58	−2.91
USA	Obese class 3	24	19.9	4.1	8.42	6.98	1.44
USA	Total	285					

Pearson Product Moment correlations between study variables (* *p* < 0.05; ** *p* < 0.01; *** *p* < 0.001). *p*/w = per week; PA = physical activity.

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
