# Peer review of "The Association between Esports Participation, Health and Physical Activity Behaviour"

_ijerph, 2020, doi:10.3390/ijerph17197329_

Round 1

Reviewer 1 Report

The study used a relatively large sample to study the descriptive nature of esport group. However, I feel that this study should not be published because of the quality of analysis.

1) My major concern is that although the total number of participants is large, they are from 63 contries, and even the 5th large contry only have 27 participants which is a very limited number. These large mixed sample may not generate indicative and imfornational results. I woul like to see all analysis with global, USA, and Australia, and others seperated, and comparision if there were any difference.

2) The results were mainly descriptive. This limited the interest of readers, and may not provide explaination of the results. The authors are encouranged to deep dig into the data, and found more important mechanism.

Minor concerns,

1) The table 1 has no title, and table 2 has a very limited title.

2) drinking and smoking behavior is very low as suggested by the results 0.3 days per week, this may suggest that there were many 0s in the data. Averaging them may not be a good idea.

Author Response

The authors wish to thank the three reviewers for taking the time to provide supportive, insightful, and constructive comments. We have endeavoured to respond to these comments in a manner that is reflective of their intentions.

1) My major concern is that although the total number of participants is large, they are from 63 contries, and even the 5th large contry only have 27 participants which is a very limited number. These large mixed sample may not generate indicative and imfornational results. I woul like to see all analysis with global, USA, and Australia, and others seperated, and comparision if there were any difference.

Thank you for this comment. We have provided a new table which indicates the number of participants obtained from the different countries the participants originated. We have also added to the limitations that there is a bias based on the language of the survey. As a number of countries only have 1 representing participant, we have opted to keep them grouped under global, only doing further analysis on countries with sufficient participants for meaningful statistical testing. However, we would like to stress that our sample is representative of esports. Hence, esports is an international endeavour which doesn’t have national boundaries. Many competitions are international in nature and players from across the world play each other. As such we believe that sampling esports players across the globe is the most appropriate method to provide a true representation of esports players. Secondly, there is currently no data available on how many individuals play esports in specific countries.

Line 126

Lines 406-410

2) The results were mainly descriptive. This limited the interest of readers, and may not provide explaination of the results. The authors are encouranged to deep dig into the data, and found more important mechanism.

We are unsure about this comment. The aim of this research was to provide information on the health and health behaviours of esports players which is a relatively new topic and as such to some extent exploratory in nature. We made some explicit predictions which were examined using appropriate statistical test. Some of the information is descriptive in nature (e.g., smoking and drinking behaviours) but that is in line with the aim of the paper. The data collected in this study are cross-sectional in nature and as such can not provide information on the underlying reasons why, for example, smoking behaviour was relatively low compared to international data. We have tried to provide some explanations in the discussion on our data but are not able to provide mechanistic explanations based on our data set.

1) The table 1 has no title, and table 2 has a very limited title.

Thank you for this comment. We have adjusted the manuscript accordingly.

Lines 227, 232

2) drinking and smoking behavior is very low as suggested by the results 0.3 days per week, this may suggest that there were many 0s in the data. Averaging them may not be a good idea.

This is the way data on smoking and drinking behaviour are collected. It is in days per week across the population. This makes comparison with other data possible. We have, however, now reported the response rate to smoking and drinking behaviour (i.e., how many did responds 0, 1 etc). in a table. We hope this provides useful information.

Line 230

Reviewer 2 Report

Thank you for the opportunity to review the manuscript entitled, "The association between esports participation, health and physical activity behaviour”.

I believe this study investigated a topic relevant to the readers of “IJERPH”.  The use of Information Technology and Communications (ICT) is an essential aspect of modern societies. Video games represent a very popular leisure activity in many countries and the participation in the esports grows rapidly worldwide. Playing and competing in esports games have significant consequences for the players’ health. There is a growing interest in research into the of health risk factors associated with the sedentary nature of Esports.

This paper is well written and follows well accepted standards of academic writing. Strengths include interest, detailed analysis and importance of the study. Some weaknesses are clearly defined in limitations. However, minor revisions may prove beneficial.

Introduction:

In relation to “BMI and population reference”, need be addressed in the introduction and study aims section. It is an objetive important in this study.

Materials and Methods:

Participants: is need the percent of males and females, the age and the average age...

The instruments must appropriate to the research questions. The instruments must always display two important qualities: reliability and validity. When a factor is defined by 4-5 items with loadings above .50, it is a solid factor with practical relevance.

Results:

The table 1 must be clearly presented. Is need the number of the participants  (N) of all variables and categories.

The MANOVA analysis requires key assumptions: independent variables, univariate normality (Test Kolmogorov-Smirnov) and homoscedasticity, the assumption of equality of variances (Test Levene)

Discussion:

Finally, the authors should consider another limitation: The participants were not randomly assigned.

Author Response

The authors wish to thank the three reviewers for taking the time to provide supportive, insightful, and constructive comments. We have endeavoured to respond to these comments in a manner that is reflective of their intentions.

Comment

Response/Revision/Rebuttal

Location in revised MS

In relation to “BMI and population reference”, need be addressed in the introduction and study aims section. It is an objetive important in this study.

Thank you for this comment. The comparison of esports players BMI, smoking and drinking behaviour as a fundamental aim of this study has been added to the abstract, introduction and aims.

Lines 14-17, 58-59, 104-106

Participants: is need the percent of males and females, the age and the average age...

Thank you for this comment. We have adjusted the manuscript accordingly.

We are not able to add participants age and average age as age data was not collected.

Lines 117-118

The instruments must appropriate to the research questions. The instruments must always display two important qualities: reliability and validity. When a factor is defined by 4-5 items with loadings above .50, it is a solid factor with practical relevance

The instruments used in this study were carefully selected based on their psychometric properties and the most parsimonious way of obtaining the relevant information. For example, the SF1 is a reliable and valid instrument to measure general health with good predictive validity. We used self-reported height and weight to calculate BMI. We are well aware of the bias in reporting this and made the required adjustment as suggested in the literature. Similarly, for PA we used an instrument which has been developed by Bull et al. (2004) for the Active Australia PA recommendations. This has been shown to be a valid and reliable correlate of objectively measured PA (Prochaska et al., 2001) and Australian PA recommendations for 12-18 year olds (DHA, 2004). The questions on smoking and drinking behaviour have been extensively used in epidemiological research and the adaptations made have been well validated and allowed for meaningful comparisons with other studies. Finally, the item to measure esports frequency was also adopted from published studies and has been validated previously (e.g., Simons, et al., 2014; Turel etal., 2017). We have now provided more information in the method section on the psychometric properties of the instruments used.

Lines 128-133, 135-137, 139-140, 153-155, 165-166, 168-170

The table 1 must be clearly presented. Is need the number of the participants  (N) of all variables and categories

Thank you for this comment. We have adjusted the manuscript accordingly.

Line 227

The MANOVA analysis requires key assumptions: independent variables, univariate normality (Test Kolmogorov-Smirnov) and homoscedasticity, the assumption of equality of variances (Test Levene)

Thank you for this comment. We checked for normality (Kolmogorov-Smirnov) and homoscedasticity (Levene). We have provided information in the revised manuscript on this. In saying this, normality tests are not really required for sample sizes over N > 40 (really, they are only needed for small sample sizes). In addition, for items like smoking and drinking and the used response options it is likely that the data are not normally distributed. Hence, individuals are more likely to not smoke or smoke regularly (e.g. 6-7 days per week) and less likely to smoke only for a few days a week. Removing outliers, an option when data are not normally distributed, would not be appropriate in such a situation.

Lines 212-215

Finally, the authors should consider another limitation: The participants were not randomly assigned.

Thank you for this feedback. It is clear that we have not been specific enough in our description of how participants were selected. The manuscript has been readjusted to more specifically discuss participant selection.

Participants self-selected to take part in the study by choosing whether or not to complete the online survey. As the study was cross-sectional and not experimental in nature, participants were not assigned in any way, but opted to self-select to take part in the study.

Lines 121-124

Reviewer 3 Report

The research objective is clear, but the development of the article is somewhat chaotic. The studio has major gaps. The method must be thoroughly reviewed. There are serious drawbacks in the sample and in the participation process. The study variables are not clear. Either the approach of the article fails or the researchers have not been able to clearly convey their research. I wish and hope that this is the second option.

  1. Materials and Methods 108

2.1. Participants: Add: number of men and women and statistical evidence of absence of gender bias Also, add mean age and standard deviation. It is not determined which ethics committee studied the feasibility of this study. It is a necessary and essential part. Do not forget to attach the code or signature that verifies this information. The distribution of the sample does not seem equitable. Please provide statistical tests where the percentage of each nation is representative for that country according to its number of inhabitants. You should also add a table explaining the place of residence of each of the participants.

2.2. Measures: On each instrument add: number of items, maximum and minimum score and information regarding the reliability of these instruments. Also briefly explain which categories are obtained. This information is partially described in some instruments. It should be in all of them.

2.3. Procedure: The type of sampling is not specified: random, stratum, etc.? It only explains how they contacted the population. Describe this process in more detail.

2.4. Analysis Strategy. Take a closer look at this point. Think about what you want to study, what tests you should do and what information you have. It is not a matter of making tests to do until you get meaningful data. Select the statistical tests well, think if they are consistent.

  1. Results

Table 1.The variables of your study according to your methodology are these:

General Health (SF-1):

Drinking and smoking:

Body Mass Index (BMI):

Physical Activity (PA):

Player in-game rank:

Gaming behaviours

Why don't all of them appear in table 1? Why does the name change in some of them? Always refer to your variables with the same terms, preferably with the name of the test.

Table 2. Homogenize the name of the variables with the tests you have used in the methodology.

There's a serious problem with the results. The sample appears to have a significant bias by nationality. If the proportions are not well adjusted, the study is meaningless.

Table 3. If the "Body Mass Index (BMI)" variable is a quantitative variable and you are comparing your sample BMI data with the NCDRisC (2017) world population BMI data, why do you do an Xi-square test? This test is used on qualitative or categorical variables. It is not logical to use this test. It is obvious that it can be done, but it is a matter of forcing the data and ignoring its nature. Added to this situation is the division by nationality, which is again a problem.

Author Response

The authors wish to thank the three reviewers for taking the time to provide supportive, insightful, and constructive comments. We have endeavoured to respond to these comments in a manner that is reflective of their intentions

Participants: Add: number of men and women and statistical evidence of absence of gender bias Also, add mean age and standard deviation. 

Thank you for this comment. We have adjusted the manuscript accordingly.

We are not able to add participants age and average age as age data was not collected.

Lines 121-124

It is not determined which ethics committee studied the feasibility of this study. It is a necessary and essential part. Do not forget to attach the code or signature that verifies this information.

Thank you for this comment. We have adjusted the manuscript to include the QUT ethics approval number.

Lines 178-179

The distribution of the sample does not seem equitable. Please provide statistical tests where the percentage of each nation is representative for that country according to its number of inhabitants.

Thank you for this comment. We have provided a new table which indicates the number of participants obtained from the different countries the participants originated. We have also added to the limitations that there is a bias based on the language of the survey. As a number of countries only have 1 representing participant, we have opted to keep them grouped under global, only doing further analysis on countries with sufficient participants for meaningful statistical testing. However, we would like to stress that our sample is representative of esports. Hence, esports is an international endeavour which doesn’t have national boundaries. Many competitions are international in nature and players from across the world play each other. As such we believe that sampling esports players across the globe is the most appropriate method to provide a true representation of esports players. Secondly, there is currently not data available on how many individuals play esports in specific countries.

Lines 126, 406-410

You should also add a table explaining the place of residence of each of the participants.

Thank you for this comment. A frequency table has been included with a list of participants country of residence frequency.

Line 126

Measures: On each instrument add: number of items, maximum and minimum score and information regarding the reliability of these instruments. Also briefly explain which categories are obtained. This information is partially described in some instruments. It should be in all of them.

Thank you for this comment. We have adjusted the description of each measure to include the number of items, the maximum and minimum scores. As each item is a single item question, chronbach’s alpha can’t be established. We have, however, provided more information on studies which have validated the instruments used in our study. Hence, we have been careful in selecting these measures based on their psychometric properties and the most parsimonious way of obtaining information from participants.

Procedure: The type of sampling is not specified: random, stratum, etc.? It only explains how they contacted the population. Describe this process in more detail.

Thank you for this feedback. It is clear that we have not been specific enough in our description of how participants were selected. The manuscript has been readjusted to more specifically discuss participant selection.

Participants self-selected to take part in the study by choosing whether or not to complete the online survey. As the study was cross-sectional and not experimental in nature, participants were not assigned in any way, but opted to self-select to take part in the study.

Lines 121-124

Analysis Strategy. Take a closer look at this point. Think about what you want to study, what tests you should do and what information you have. It is not a matter of making tests to do until you get meaningful data. Select the statistical tests well, think if they are consistent.

Thank you for this comment, to clarify more clearly the link between analysis strategy and the aims for the study we have included a link between the study aims and the specific analysis methods used to determine the results.

Lines 191-205

Table 1.The variables of your study according to your methodology are these:

General Health (SF-1):

Drinking and smoking:

Body Mass Index (BMI):

Physical Activity (PA):

Player in-game rank:

Gaming behaviours

Why don't all of them appear in table 1? Why does the name change in some of them? Always refer to your variables with the same terms, preferably with the name of the test.

We have double checked the paper, an believe we have detected and changed all cases of incorrectly labelled variable names. Thank-you for pointing this out. The manuscript has been changed to reflect this point. We have also changed gaming behaviours to esports frequency, to better reflect the question.

Lines 88, 98-99, 104, 110, 136, 219, 221-222, 227, 232, 249-255, 269, 274, 279, 288297-298, 338-339, 365, 367, 376, 385, 392, 412-413-415, 417-418

There's a serious problem with the results. The sample appears to have a significant bias by nationality. If the proportions are not well adjusted, the study is meaningless

Thank you for this comment. We have provided a new table which indicates the number of participants obtained from the different countries the participants originated. We have also added to the limitations that there is a bias based on the language of the survey. As a number of countries only have 1 representing participant, we have opted to keep them grouped under global, only doing further analysis on countries with sufficient participants for meaningful statistical testing. However, we would like to stress that our sample is representative of esports. Hence, esports is an international endeavour which doesn’t have national boundaries. Many competitions are international in nature and players from across the world play each other. As such we believe that sampling esports players across the globe is the most appropriate method to provide a true representation of esports players. Secondly, there is currently not data available on how many individuals play esports in specific countries.

Lines 126, 460-410

Table 3. If the "Body Mass Index (BMI)" variable is a quantitative variable and you are comparing your sample BMI data with the NCDRisC (2017) world population BMI data, why do you do an Xi-square test? This test is used on qualitative or categorical variables. It is not logical to use this test. It is obvious that it can be done, but it is a matter of forcing the data and ignoring its nature. Added to this situation is the division by nationality, which is again a problem.

We appreciate this feedback. The data supplied by the NCDRisC was in the form of percentages, not raw data. It was for this reason we used a Chi-square test. If we had the raw data, which NCDRisC couldn’t supply than we would have opted for a different method of data analysis.

Regarding the nationality issue, as noted previously esports is a global activity in which players regularly play against players from various countries. There is also no existing data about the prevalence of esports players in different countries for us to use as reference data, making it impossible to determine if our sample is skewed compared to individual country reference data.

Round 2

Reviewer 1 Report

None

Reviewer 3 Report

Dear Authors and Publisher,
I appreciate your modifications. I am aware of the effort you have made, as these were important and profound. However, the procedure has not been the right one. The methodological errors were made at the beginning of the investigation and have weighed it down. In my opinion it is not publishable despite its changes. However, I will leave the decision to the editor and the other reviewers.
I am aware that the next investigation will be carried out in an appropriate manner. I regret the situation and I empathize with you.